

# Investigation of the effects of estrogen on skeletal gene expression during zebrafish larval head development

Ehsan Pashay Ahi[1], Benjamin S. Walker[2], Christopher S. Lassiter[2] and Zophonías O. Jónsson[1,3]

[1] Institute of Life and Environmental Sciences, University of Iceland, Reykjavik, Iceland
[2] Biology Department, Roanoke College, Salem, VA, United States
[3] Biomedical Center, University of Iceland, Reykjavik, Iceland

## ABSTRACT

The development of craniofacial skeletal structures requires well-orchestrated tissue interactions controlled by distinct molecular signals. Disruptions in normal function of these molecular signals have been associated with a wide range of craniofacial malformations. A pathway mediated by estrogens is one of those molecular signals that plays role in formation of bone and cartilage including craniofacial skeletogenesis. Studies in zebrafish have shown that while higher concentrations of 17-$\beta$ estradiol ($E_2$) cause severe craniofacial defects, treatment with lower concentrations result in subtle changes in head morphology characterized with shorter snouts and flatter faces. The molecular basis for these morphological changes, particularly the subtle skeletal effects mediated by lower $E_2$ concentrations, remains unexplored. In the present study we address these effects at a molecular level by quantitative expression analysis of sets of candidate genes in developing heads of zebrafish larvae treated with two different $E_2$ concentrations. To this end, we first validated three suitable reference genes, *ppia2*, *rpl8* and *tbp*, to permit sensitive quantitative real-time PCR analysis. Next, we profiled the expression of 28 skeletogenesis-associated genes that potentially respond to estrogen signals and play role in craniofacial development. We found $E_2$ mediated differential expression of genes involved in extracellular matrix (ECM) remodelling, *mmp2/9/13*, *sparc* and *timp2a*, as well as components of skeletogenic pathways, *bmp2a*, *erf*, *ptch1/2*, *rankl*, *rarab* and *sfrp1a*. Furthermore, we identified a co-expressed network of genes, including *cpn1*, *dnajc3*, *esr1*, *lman1*, *rrbp1a*, *ssr1* and *tram1* with a stronger inductive response to a lower dose of $E_2$ during larval head development.

## INTRODUCTION

Craniofacial development is a critical part of embryogenesis and identification of molecular mechanisms underlying this process is important in gaining a better understanding of morphological diversity in vertebrates (*Szabo-Rogers et al., 2010*) as well as human health (*Oginni & Adenekan, 2012*). The viscerocranium in humans is of interest because of

Corresponding author
Ehsan Pashay Ahi, epa1@hi.is

oro-facial clefts and associated malformations (*Marazita, 2012*). The vertebrate craniofacial skeleton, including the viscerocranium, is built from neural-crest derived tissues. Changes in these tissues over evolutionary time have given rise to a wide diversity of facial morphologies among vertebrate species (*Trainor, Melton & Manzanares, 2003*; *Bronner & LeDouarin, 2012*).

Estrogens, steroid hormones synthesized by aromatase from androgenic precursors, have recently been shown to affect craniofacial development (*Fushimi et al., 2009*; *Cohen et al., 2014*). Though estrogens are normally thought of as sex hormones, they affect a variety of tissues including the cardiovascular and skeletal systems (*Hall, Couse & Korach, 2001*; *Allgood et al., 2013*; *Cohen et al., 2014*). Estrogens signal through classical nuclear receptors (ER-alpha and ER-beta) (*Jia, Dahlman-Wright & Gustafsson, 2015*) and a G-protein coupled receptor, GPR-30 (*Jenei-Lanzl et al., 2010*). These signaling proteins are found in the chondrocytes of many vertebrate species (*Tankó et al., 2008*) and are present during chondrogenesis (*Jenei-Lanzl et al., 2010*; *Elbaradie et al., 2013*). Among vertebrates, teleost fish are highly diversified in craniofacial morphology and estrogen has been shown to greatly affect chondrogenesis in many of the fish species studied so far, including tilapia, fathead minnow, and zebrafish (*Ng, Datuin & Bern, 2001*; *Warner & Jenkins, 2007*; *Cohen et al., 2014*). Furthermore, the teleost zebrafish is a well-developed model system for studying both embryonic development and human disease and it can be useful in understanding the development of the human viscerocranial skeleton as these processes are well-conserved among vertebrates (*Kuratani, Matsuo & Aizawa, 1997*).

Low concentrations of estrogen cause subtle changes in craniofacial morphogenesis during zebrafish larval development (*Cohen et al., 2014*). These changes are characterized by a shorter snout, flatter face and wider angles of cartilage elements in the viscerocranium (*Cohen et al., 2014*). Another recent study has also demonstrated that an estrogen mediated signal underlies the evolution of shorter snouts and flatter faces in females of some reptilian species (*Sanger et al., 2014*). These observations suggest that similar mechanisms might underpin hormone-based phenotypic plasticity and diversity (*Dufty, 2002*), as well as subtle differences in head/skeletal morphology of dimorphic sexes (*Loth & Henneberg, 2001*; *Fujita et al., 2004*; *Callewaert et al., 2010*). Therefore, studies offering insights into molecular mechanisms rendering the observed phenotypes caused by hormonal changes would be called for.

The subtle changes in craniofacial skeletogenesis mediated by low concentrations of Estradiol ($E_2$) are likely to be a result of differences in level and timing of the expression of skeletogenesis-associated genes during head development (*Albertson et al., 2010*; *Ahi et al., 2014*; *Gunter, Koppermann & Meyer, 2014*; *Powder et al., 2015*). These morphological changes were only revealed by careful measurements of skeletal elements at zebrafish larval stages (*Cohen et al., 2014*), therefore the identification of responsible genes might also require precise expression studies in developing heads of zebrafish larvae using a sensitive tool such as quantitative real-time PCR (qPCR) (*Bustin, 2000*; *Kubista et al., 2006*). In the present study, we set out to quantitatively assess the effects of estrogen on the expression dynamics of candidate genes which are known as potential targets of estrogen pathway and also involved in craniofacial skeletogenesis in different vertebrate species (Table 1). We

**Table 1** Selected putative estrogen-regulated candidate genes, and available literature indicating their role in craniofacial development/skeletal formation in zebrafish or other vertebrates.

| Gene symbol | Related function | Viscerocranial expression during zebrafish development | Potential estrogen responsive[*] | Craniofacial skeletogenesis | | References |
|---|---|---|---|---|---|---|
| | | | | Shortened snout[**] | Other effects | |
| alx4 | Patterning and development of craniofacial skeleton | + | + | + | + | (Qu et al., 1999; Joshi, Chang & Hamel, 2006; Lours-Calet et al., 2014) |
| bmp2 (a/b) | Induction of bone and cartilage formation | + | + | ? | + | Thisse et al. (2001), Thisse et al. (2004), Zhou et al. (2003), Nie, Luukko & Kettunen (2006), Hu, Colnot & Marcucio (2008), and Yamamoto, Saatcioglu & Matsuda (2013) |
| col2a1a | Extracellular matrix formation in cartilaginous tissues | + | + | + | + | Maddox et al. (1997), Eames et al. (2010) and Maneix et al. (2014) |
| ctsk | Bone remodelling and resorption | + | + | + | + | Thisse et al. (2004), Troen (2006), Petrey et al. (2012) and Ahi et al. (2014) |
| dlk1 | Differentiation of skeletal cells | ? | + | + | + | Abdallah et al. (2011) |
| erf | Regulation of cellular senescence | ? | + | + | + | Frasor et al. (2003) and Twigg et al. (2013) |
| esrra | Regulation of estrogen mediated pathway | + | + | ? | + | Bonnelye & Aubin (2005), Bonnelye et al. (2007) and Auld et al. (2012) |
| esr1 | A ligand-activated receptor for estrogen | ? | + | ? | + | O'Lone et al. (2004) and Syed et al. (2005) |
| ets2 | Regulation of developmental genes and apoptosis | ? | + | + | + | Sumarsono et al. (1996), Deblois et al. (2009) and Ahi et al. (2014) |
| mmp (2/9/13) | Extracellular matrix formation and signal transduction | + | + | + | + | Breckon et al. (1999), Tüshaus et al. (2003), Marin-Castaño et al. (2003), Lu et al. (2006), Hillegass et al. (2007a), Hillegass et al. (2007b), Mosig et al. (2007) and Nilsson, Garvin & Dabrosin (2007) |
| opg | Negative regulation of bone resorption | ? | + | ? | + | Bord et al. (2003) and Whyte et al. (2002) |
| pbx1 (a/b) | Co-ordination of chondrocyte proliferation and differentiation | + | + | + | + | Selleri et al. (2001), Thisse & Thisse (2005) and Magnani et al. (2011) |
| ptch (1/2) | Receptors for hedgehog signalling pathway | + | + | + | + | (Fushimi et al., 2009; Roberts et al., 2011) |

**Table 1** (*continued*)

| Gene symbol | Related function | Viscerocranial expression during zebrafish development | Potential estrogen responsive[*] | Craniofacial skeletogenesis | | References |
|---|---|---|---|---|---|---|
| | | | | Shortened snout[**] | Other effects | |
| *rankl* | Osteoclast differentiation and activation | ? | + | ? | + | *Bord et al. (2003)* and *Lézot et al. (2015)* |
| *rarab* | A receptor for retinoic acid signalling pathway | + | + | + | + | (*Lohnes et al., 1994*; *O'Lone et al., 2004*; *Linville et al., 2009*) |
| *runx2b* | Osteoblast differentiation and skeletal morphogenesis | + | + | + | + | *Sears et al. (2007)*, *McCarthy et al. (2003)* and *Flores et al. (2006)* |
| *sfrp1a* | A soluble modulator of Wnt signalling pathway | ? | + | + | + | *Satoh et al. (2006)*, *Trevant et al. (2008)*, *Yokota et al. (2008)*, *Fukuhara et al. (2013)* and *Ahi et al. (2014)* |
| *Shh* (a/b) | Activators of hedgehog signalling pathway | + | ? | ? | + | *Hu & Helms (1999)* and *Swartz et al. (2012)* |
| *sox9b* | Chondrocyte differentiation | + | + | ? | + | *Yan et al. (2005)*, *Bonnelye et al. (2007)* and *Lee & Saint-Jeannet (2011)* |
| *sparc* | Extracellular matrix synthesis and regulation of cell growth | + | + | + | + | *Lehane et al. (1999)*, *Renn et al. (2006)* and *Rotllant et al. (2008)* |
| *spp1* | Attachment of osteoclasts to ECM in bone | + | + | + | + | *Craig & Denhardt (1991)*, *Vanacker et al. (1998)* and *Venkatesh et al. (2014)* |
| *timp2a* | Inhibition of mmps and regulation of tissue homeostasis | ? | + | + | ? | *Dew et al. (2000)*, *Lam et al. (2009)*, *Letra et al. (2012)*, *Wang & Ma (2012)* and *Ahi et al. (2014)* |

**Notes.**

[*]The estrogen responsiveness indicates either transcriptional regulation or transactivation and the related information are mainly obtained from different model vertebrates, such as human and mouse, than teleost fishes.

[**]The shortened snout indicates the skeletal effects resulted from decrease in the length or changes in morphology of viscerocranial skeletal elements in different vertebrate species. This could bear a resemblance to an estrogen mediated shorter snout and flatter face phenotype in zebrafish.

hypothesized that these genes may be critical to the estrogen modulation of craniofacial skeletogenesis. We first identified the most stably expressed reference genes in developing heads of zebrafish treated with two doses of estrogen ($2\,\mu M$ and $5\,\mu M$) across five stages in larval development. Then, we accurately measured small changes in the expression levels of the candidate genes. In addition, we have used available co-expression data from zebrafish to identify a co-expressed network of genes with greater transcriptional response to the lower dose of estrogen ($2\,\mu M$) during larval head development.

## METHODS

### Fish husbandry, treatment and sampling

Adult zebrafish were fed a diet of live brine shrimp supplemented with Ziegler zebrafish diet (Pentair) and maintained on a 14/10 day/night cycle. Embryos were raised in E3B (5 mM NaCl, 0.17 mM KCl, 0.33 mM $CaCl_2$, 0.33 mM $MgSO_4$, 0.00025% methylene blue). Embryos were treated with estrogen (17$\beta$-estradiol, $E_2$, Sigma) dissolved in ethanol and diluted in E3B for a final ethanol concentration of 0.1%. Control fish were treated with 0.1% ethanol with no developmental malformations as described previously (*Cohen et al., 2014*). For each treatment group (estrogen concentration), zebrafish larva were raised in Petri dishes, and treatment solutions were refreshed daily until the stages indicated (3, 4, 5, 6 and 7 days post fertilization, dpf). Three biological replicates of 30 larva were collected at each time-point (3–7 dpf) and for each treatment group (control, 2 μM $E_2$, and 5 μM $E_2$) for a total of 90 larva at each time-point and treatment. The fishes were anesthetized with 0.4% tricaine (MS-222, Sigma). Isolated heads (anterior to the yolk sac) were placed into RNAlater (Qiagen) and stored frozen until RNA isolation. Zebrafish experiments were performed under the Roanoke College IRB protocol #14BIO76.

### RNA isolation and cDNA synthesis

Around 30 heads of zebrafish from each treatment group and larval stage were pooled in TRI Reagent (Sigma) and homogenized with a disposable Kontes Pellet Pestle Cordless Motor tissue grinder (Kimble Kontes). RNA was prepared according to manufacturer's instructions and dissolved in 50 μl RNase-free water. RNA samples were treated with DNase (New England Biolabs) to remove contaminating DNA. Quantity of the resulting RNA samples was assessed using a NanoDrop ND-1000 UV/Vis-Spectrophotometer (NanoDrop Technologies). The quality of the RNA samples was evaluated by agarose gel electrophoresis and all samples displayed intact 28 S and 18 S rRNA without noticable high molecular weight genomic DNA contamination. cDNA was prepared from 1000 ng of RNA using the High capacity cDNA Reverse Transcription kit (Applied Biosystems), according to manufacturer's protocol. Several samples without addition of reverse transcriptase (-RT samples) were prepared to confirm the absence of genomic DNA. cDNA was diluted 3 fold in water for further use in quantitative real-time PCR.

### Gene selection, Primer design and real-time qPCR

In order to validate suitable reference genes for accurate measurement of the transcriptional changes of candidate genes by qPCR, we selected 7 potential reference genes based on published studies in zebrafish (Table S1) (*McCurley & Callard, 2008*; *Pelayo et al., 2012*; *Schiller et al., 2013*), none of which have been validated during development or in zebrafish head. In addition we selected 28 target genes that are known as potential targets of the estrogen pathway in different vertebrate species, and also involved in craniofacial skeletal formation/morphogenesis (Table 1 and Table S1). Finally, we extended our list of candidates by adding more genes showing co-expression with the estrogen receptor *esr1* based on the zebrafish database COXPRESDb (http://coxpresdb.jp/) version 6.0 (*Obayashi & Kinoshita, 2011*). To obtain the maximum number of coexpressed genes with a high degree of

reliability, we filtered the genes by setting the mutual rank (MR) to the top-ranked 2000 and the Supportability score of minimum 1 (as described by *Obayashi & Kinoshita, 2011*). This yielded 338 candidate genes, and from them, we selected 11 genes with reported craniofacial expression during zebrafish development according to the ZFIN database (http://zfin.org) (*Bradford et al., 2011*) (Table S1).

Locations overlapping exon boundaries of the genes in zebrafish were determined by NCBI Spidey software (www.ncbi.nlm.nih.gov/spidey) and annotated genome sequences in the Ensembl database (http://www.ensembl.org/Danio_rerio). The qPCR Primers were designed on exon boundaries using Primer Express 3.0 software (Applied Biosystems, Foster City, CA, USA) and checked for self-annealing, hetero-dimers and hairpin structures with OligoAnalyzer 3.1 (Integrated DNA Technology) (Table S1).

Real-time PCR was performed in 96 well-PCR plates on an ABI 7500 real-time PCR System (Applied Biosystems) using Maxima SYBR Green/ROX qPCR Master Mix (2X) as recommended by the manufacturer (Thermo Fisher Scientific, St Leon-Rot, Germany). Each biological replicate was run in duplicate together with no-template control (NTC) in each run for each gene and experimental set-up per run followed the preferred sample maximization method (*Hellemans et al., 2007*). The qPCR was run with a 2 min hold at 50 °C and a 10 min hot start at 95 °C followed by the amplification step for 40 cycles of 15 sec denaturation at 95 °C and 1 min annealing/extension at 60 °C. A dissociation step (60 °C–95 °C) was performed at the end of the amplification phase to identify a single, specific product for each primer set (Table S1). Primer efficiency values (E) were calculated with the LinRegPCR v11.0 programme (http://LinRegPCR.nl) (*Ramakers et al., 2003*) analysing the background-corrected fluorescence data from the exponential phase of PCR amplification for each primer-pair and those with E less than 0.9 were discarded and new primers designed (Table S1).

## Data analysis

To detect the most stably expressed reference genes, three ranking algorithms; BestKeeper (*Pfaffl et al., 2004*), NormFinder (*Andersen, Jensen & Ørntoft, 2004*) and geNorm (*Vandesompele et al., 2002*) were employed. The standard deviation (SD) based on Cq values of the larval stages and treatment groups was calculated by BestKeeper to determine the expression variation for each reference gene. In addition, BestKeeper determines the stability of reference genes based on correlation to other candidates through calculation of BestKeeper index (r). GeNorm measures mean pairwise variation between each gene and other candidates, the expression stability or *M* value, and it excludes the gene with the highest *M* value (least stable) from subsequent analysis in a stepwise manner. Moreover, geNorm determines the optimal number of reference genes through calculation of pairwise variation coefficient ($Vn/n + 1$) between two sequential normalisation factors ($NFn$ and $NFn + 1$) and extra reference genes are added until the variation drops below the recommended threshold of 0.15 (*Vandesompele et al., 2002*). NormFinder identifies the most stable genes (lowest expression stability values) based on analysis of the sample subgroups (stage and treatment group) and estimation of inter- and intra-group variation in expression levels.
For the analysis of qPCR data, the difference between $Cq$ values ($\Delta Cq$) of the reference genes and the target genes was calculated for each gene; $\Delta Cq_{target} = Cq_{target} - Cq_{reference}$. The geometric mean of Cq values of three best ranked reference genes, *ppia2*, *rpl8* and *tbp* (see the ranking algorithms above), was used as Cq $_{reference}$ in the $\Delta Cq$ calculations. All samples were then normalized to the $\Delta Cq$ value of a calibrator sample to obtain a $\Delta\Delta Cq$ value ($\Delta Cq_{target} - \Delta Cq_{calibrator}$). For each primer pair a biological replicate in the control group at 3dpf was selected as the calibrator sample. Relative expression quantities (RQ) were calculated based on the expression level of the calibrator sample ($E^{-\Delta\Delta Cq}$) (*Pfaffl, 2001*). The RQ values were then transformed to logarithmic base 2 values (or fold differences; FD) for statistical analysis (*Bergkvist et al., 2010*). A two-way analysis of variance (ANOVA) followed by post hoc Tukey's honest significant difference (HSD) test was implemented for each reference or target gene with larval stages and treatment groups as categorical variables. To assess similarities in expression patterns of the genes Pearson correlation coefficients ($r$) were calculated for all gene pairs using the data from 3 treatments at 5 larval stages (degree of freedom = 13). R (http://www.r-project.org) was used for all statistical analysis.

# RESULTS

## *tbp*, *ppia2* and *rpl8* are the most suitable reference genes

Real-time quantitative PCR for the 7 reference gene candidates was performed on cDNA generated from zebrafish head homogenates in three treatment groups at five larval stages. The expression levels of the candidates varied from *ppia2*, with the highest expression (lowest $Cq$) (Fig. 1A), to *tbp* with the lowest expression (highest $Cq$). Statistical analysis revealed that all of the candidates except *actb1* are stably expressed between the treatment groups (Fig. 1B). However, only *tbp* showed constant expression during the larval stages examined. Two genes, *ppia2* and *rpl8*, were also stably expressed in developing heads of zebrafish larvae except for the first stage (3dpf). Based on these results *tbp* followed by *ppia2* and *rpl8* were found to be the overall most stable reference genes both over time and between the treatment groups. The candidate reference genes were ranked using three algorithms, i.e., BestKeeper, geNorm and NormFinder, and based on standard deviation (SD) as described in *Ahi et al. (2013)* (Table 2). In all of the analyses three genes; *ppia2*, *rpl8* and *tbp*, were the three highest ranking candidates, however their order varied between the rankings (Table 2). Furthermore, geNorm suggested the use of the three best ranked candidate genes as sufficient for accurate normalisation (Fig. S1). The data reflect the high expression stability of the best ranked candidate genes and suggests the combination of *ppia2*, *rpl8* and *tbp* as a suitable and sufficient normalization factor to accurately quantify small differences in gene expression in developing heads of zebrafish larvae across the $E_2$ treatment groups.

## Components of different signalling pathways and skeletogenesis-associated genes are affected by estrogen during larval head development

The selected 28 candidate target genes, listed in Table 1, can be classified into distinct functional groups; (I) estrogen receptors with potential involvement in vertebrate

**A**

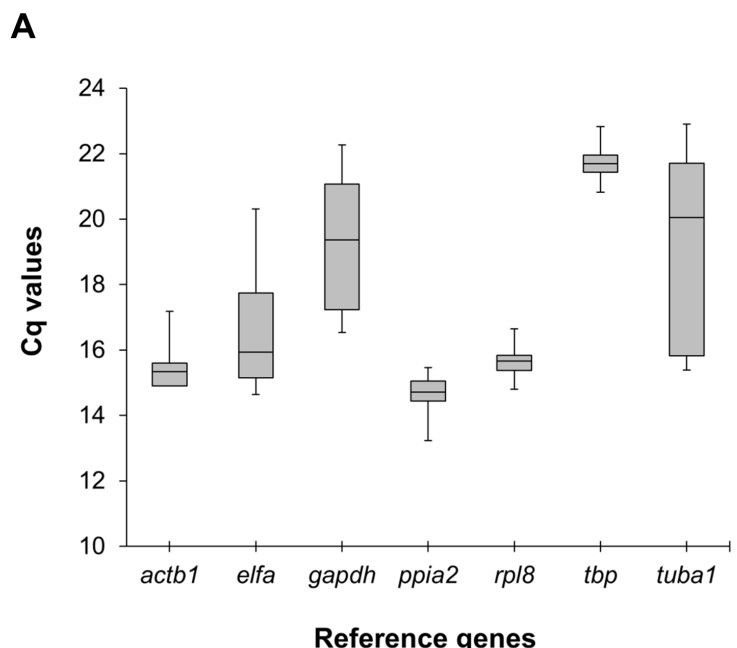

**B**

| Gene | ANOVA treatments | HSD treatments | | | ANOVA larval stage | HSD larval stage | | | | |
|---|---|---|---|---|---|---|---|---|---|---|
| actb1 | P = 0.039 | Ctl | 2µM | 5µM | P = 9.77e-08 | 3dpf | 4dpf | 5dpf | 6dpf | 7dpf |
| elfa | P = 0.758 | NS | | | P = 3.44e-06 | 3dpf | 4dpf | 5dpf | 6dpf | 7dp |
| gapdh | P = 0.437 | NS | | | P = 0.002 | 3dpf | 4dpf | 5dpf | 6dpf | 7dpf |
| ppia2 | P = 0.64 | NS | | | P = 0.001 | 3dpf | 4dpf | 5dpf | 6dpf | 7dpf |
| rpl8 | P = 0.084 | NS | | | P = 2.04e-05 | 3dpf | 4dpf | 5dpf | 6dpf | 7dpf |
| tbp | P = 0.532 | NS | | | P = 0.752 | NS | | | | |
| tuba1 | P = 0.502 | NS | | | P = 1.4e-13 | 3dpf | 4dpf | 5dpf | 6dpf | 7dpf |

**Figure 1** **Expression analysis of candidate reference genes in developing heads of zebrafish larvae across control and $E_2$ treated groups.** (A) Expression profiles of candidate reference genes in raw Cq values for all samples (3 treatments for each of 5 larval stages and with 3 biological replicates). The middle line denotes the median and boxes indicate the 25/75 percentiles. (B) Expression differences of candidate reference genes in the head of zebrafish during the larval development and three $E_2$ treatment groups. Fold changes in expression calculated from the qPCR data, were subjected to ANOVA and Tukey's HSD analysis to test the expression differences amongst three treatment groups (control, 2 µM and 5 µM) and across five larval stages (3 to 7dpf). White boxes represent low expression, while black boxes represent high expression. Two or more steps of shade differences in the boxes represent significantly different expression between the samples (alpha = 0.05). NS, not significant.

craniofacial development (*esrra* and *esr1*); (II) components of hedgehog (Hh) signaling pathway (*ptch1/2* and *shha/b*); (III) potential skeletogenic targets of estrogen pathway with critical roles in viscerocranial development/morphogenesis (*bmp2a/b*, *opg*, *rankl*, *runx2b* and *sox9b*); (IV) potential targets of estrogen pathway involved in ECM formation and

**Table 2   Ranking and statistical analyses of candidate reference genes using BestKeeper, geNorm and NormFinder.**

| BestKeeper | | | | geNorm | | NormFinder | |
|---|---|---|---|---|---|---|---|
| Ranking | r | Ranking | SD | Ranking | M | Ranking | SV |
| *rpl8* | 0.908 | *tbp* | 0.294 | *ppia2* | 0.111 | *rpl8* | 0.137 |
| *tbp* | 0.863 | *rpl8* | 0.343 | *rpl8* | 0.125 | *ppia2* | 0.154 |
| *ppia2* | 0.862 | *ppia2* | 0.350 | *tbp* | 0.133 | *tbp* | 0.157 |
| *actb1* | 0.687 | *actb1* | 0.396 | *actb1* | 0.26 | *actb1* | 0.287 |
| *efl1a* | 0.331 | *efl1a* | 1.358 | *efl1a* | 0.739 | *efl1a* | 1.128 |
| *tuba1* | 0.201 | *gapdh* | 1.690 | *gapdh* | 1.084 | *gapdh* | 1.382 |
| *gapdh* | 0.148 | *tuba1* | 2.773 | *tuba1* | 1.482 | *tuba1* | 2.088 |

**Notes.**

Abbreviations: SD, Standard deviation; *r*, Pearson product-moment correlation coefficient; SV, Stability value; *M*, *M* value of stability.

associated with shortened snout morphogenesis in vertebrates (*col2a1a*, *ctsk*, *mmp2/9/13*, *sparc*, *spp1* and *timp2a*); and (V) other potential targets of estrogen pathways with diverse functions which are also involved in viscerocranial skeletogenesis (*alx4*, *dlk1*, *erf*, *ets2*, *pbx1a/b*, *rarab* and *sfrp1a*). The expression levels of all candidates were measured in the three treatment groups during larval head development (Figs. 2, 3, 4 and 5). We found effects of different $E_2$ concentrations on the expression of most of the target genes, except *col2a1a* and *pbx1a*, the effects, however, were highly variable among the genes (Figs. 2, 3, 4 and 5). For instance, while some genes, i.e., *esr1*, *ptch1/2* and *rarab* displayed differential expression between the treatment groups at most of the larval stages, other genes such as *alx4*, *bmp2b*, *ctsk*, *ets2*, *opg*, etc., showed expression differences at only one stage. Among the more highly affected genes, *erf*, *esrra*, *mmp9*, *rankl*, *shha*, *sfrp1a*, *sparc* and *timp2a* were differentially expressed in at least three larval stages (Figs. 2, 3, 4 and 5). Although significant, most differences in expression levels of the target genes were slight between the treatment groups (RQ < 0.5), except for *esr1* at the last larval stages (Fig. 2). Moreover, for all of the affected genes, except *esr1* and *mmp13*, the different $E_2$ treatments had mainly repressive effects on transcription. These repressive effects were not, however, increased by higher $E_2$ concentration particularly at the last two stages when the lower $E_2$ dose (2 µM) repressed expression of many of the genes more than the higher dose. At the last three stages, the expression of *esr1* was induced at highest levels for 2 µM treatment groups (Fig. 2). The transcriptional repression by $E_2$ was also variable between the genes and it was more pronounced for *erf* and *ptch2* showing higher expression in control groups than both $E_2$ treated groups at three larval stages. Taken together, these results show significant effects of low $E_2$ concentrations on the expression of a variety of genes involved in skeletogenesis and/or craniofacial development.

We calculated the Pearson's correlation coefficient of the expression levels for the target genes over all treatment groups and larval stages and found positive expression correlation between many pairs of target genes (blue shadings in Fig. S2). Some of the genes i.e., *mmp9*, *ptch1*, *rarab* and *timp2a* displayed positive expression correlation with most of the genes whereas others such as *mmp13*, *sfrp1a*, *shhb* and *sparc* showed the least

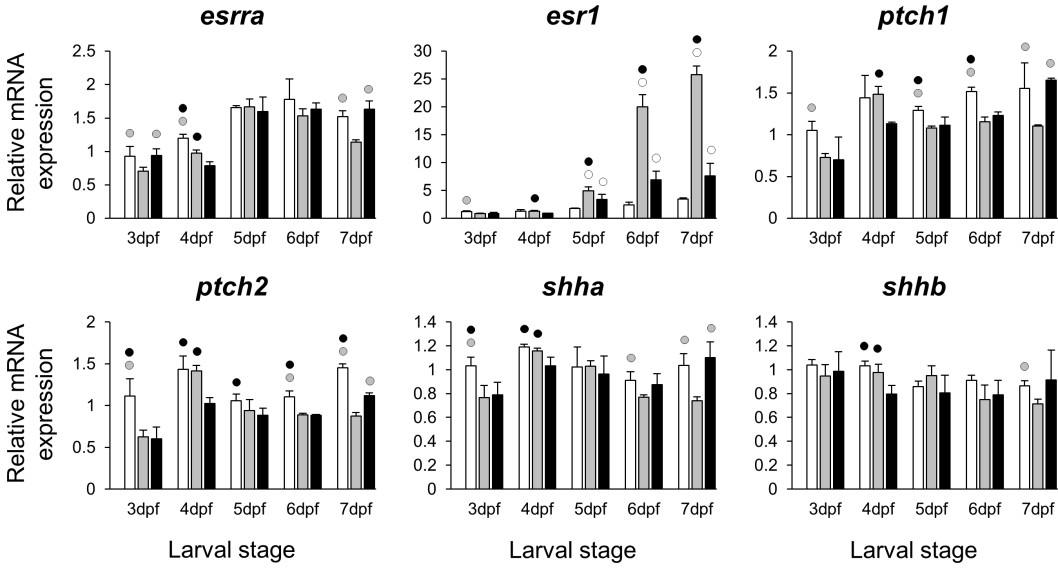

**Figure 2** **Expression differences of two estrogen receptors and components of hedgehog signaling pathway in developing heads of zebrafish larvae across control and $E_2$ treated groups.** Expression of *esrra*, *esr1*, *ptch1*, *ptch2*, *shha* and *shhb* was examined with qPCR and normalised using three highest ranked reference genes (*ppia2*, *rpl8* and *tbp*). For analysis of relative expression levels for each target gene a replicate of the control group at 3dpf was set to one. The white, grey, and black bars in each graph represent expression levels for control, 2 μM $E_2$ treated and 5 μM $E_2$ treated groups respectively. Statistical differences of each treatment group versus the others are shown in white, grey, and black circles representing higher expressed than control, 2 μM $E_2$ treated and 5 μM $E_2$ treated groups respectively ($P < 0.05$). Error bars represent standard deviation calculated from three biological replicates. Each biological replicate is from a homogenate of 30 heads.

correlated expression. Negatively correlated expression was only seen between *esr1* and *sfrp1a*, and between *shhb* and six genes including *esr1*, *ets2*, *mmp13*, *opg*, *pbx1a* and *spp1* (red shadings in Fig. S2).

## A co-expressed network of genes shows higher expression induction in lower $E_2$ treatment groups

The stronger transcriptional response of *esr1* to the lower $E_2$ treatment (Fig. 2) could indicate a distinct regulatory mechanism associated with slight increase in estrogen concentration during zebrafish larval head development. In order to identify additional genes showing similar expression dynamics, we selected 11 candidate genes constructing a co-expression network with *esr1* using co-expression data for zebrafish in COXPRESdb (*Obayashi & Kinoshita, 2011*) (Table S2). These candidates are also known to have craniofacial skeletal expression during zebrafish development based on data submitted to the ZFIN database (http://zfin.org) (*Bradford et al., 2011*). Strikingly, we found stronger inductive effects of the lower $E_2$ concentration on the expression of six genes, i.e., *cpn1*, *dnajc3*, *lman1*, *rrbp1a*, *ssr1* and *tram1* (Fig. 6). The expression of these six genes followed a similar pattern and their higher expression levels were more pronounced at the last three stages of 2 μM treatment groups. Moreover, the gene showing strongest coexpression relationship with *esr1* among the candidates, *rrbp1a*, had shown higher expression levels

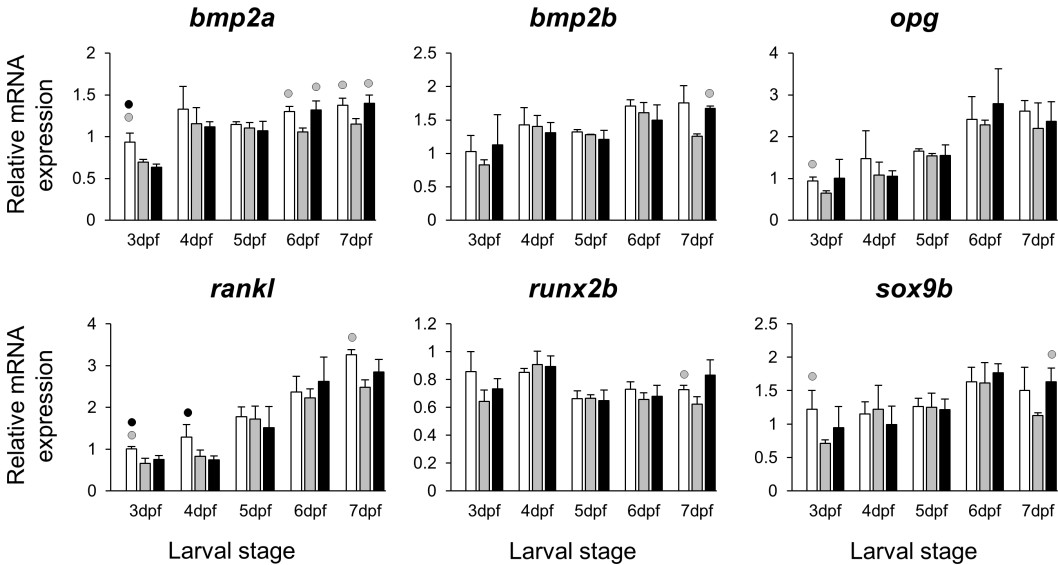

**Figure 3** **Expression differences of six potential skeletogenic targets of estrogen pathway in developing heads of zebrafish larvae across control and $E_2$ treated groups.** Expression of *bmp2a*, *bmp2b*, *opg*, *rankl*, *runx2b* and *sox9b* was examined with qPCR and normalised using three highest ranked reference genes (*ppia2*, *rpl8* and *tbp*). For analysis of relative expression levels for each target gene a replicate of the control group at 3dpf was set to one. The white, grey, and black bars in each graph represent expression levels for control, 2 μM $E_2$ treated and 5 μM $E_2$ treated groups respectively. Statistical differences of each treatment group versus the others are shown in white, grey, and black circles representing higher expressed than control , 2 μM $E_2$ treated and 5 μM $E_2$ treated groups respectively ($P < 0.05$). Error bars represent standard deviation calculated from three biological replicates. Each biological replicate was made from a homogenate of 30 heads.

at the last four stages of 2 μM treatment groups (Table S2 and Fig. 6). Finally, we also demonstrated positive expression correlations between the six candidates and *esr1*, but not the rest of the non-differentially expressed genes (blue shadings in Fig. S3).

## DISCUSSION

Estrogen signaling, through both canonical nuclear estrogen receptors and G-protein coupled receptors, is important in embryonic development (*Griffin et al., 2013*; *Shi et al., 2013*). Estrogens can act at autocrine, paracrine, and endocrine distances in the embryo and the adult (*Boon, Chow & Simpson, 2010*). Aromatase, the enzyme that synthesizes estrogens, is present in the developing brain of many species, including zebrafish (*Lassiter & Linney, 2007*) and would be a local source of the hormone during head development. In fact, the teleost brain produces relatively high levels of estrogen compared to other vertebrates (*Forlano et al., 2001*). Estrogens are thus present in the cranium of developing embryos and modulate viscerocranial development (*Fushimi et al., 2009*; *Marquez Hernandez et al., 2011*; *Cohen et al., 2014*). Estrogen signalling has been implicated in the sexual dimorphism of cranial bones in the Anolis lizard (*Sanger et al., 2014*). Hence, it may play a role in craniofacial morphological divergence among species and within sexes of the same species.

A previous attempt to identify mechanisms underlying the effects of estrogen on zebrafish craniofacial development was conducted with a high concentration of 17-$\beta$

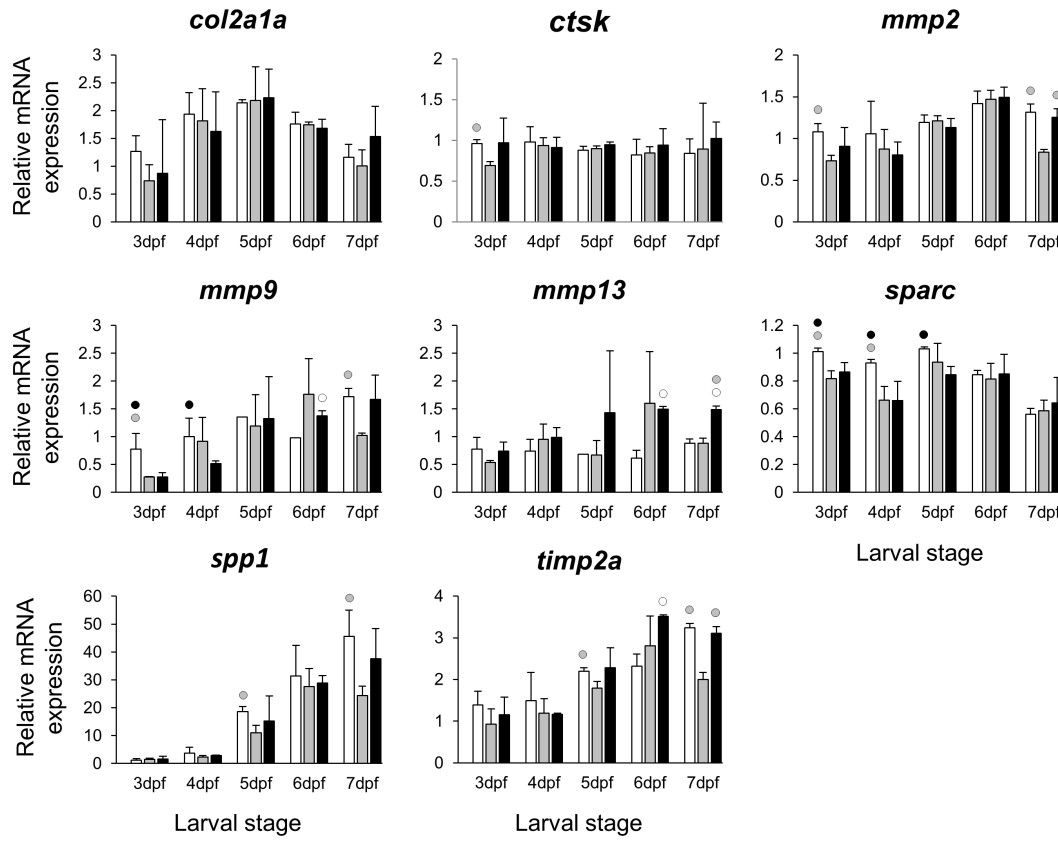

**Figure 4** Expression differences of eight potential targets of estrogen pathway involved in skeletal ECM formation examined during zebrafish larval head development across control and $E_2$ treated groups. Expression of *col2a1a*, *ctsk*, *mmp2*, *mmp9*, *mmp13*, *sparc*, *spp1* and *timp2* was examined with qPCR and normalised using three highest ranked reference genes (*ppia2*, *rpl8* and *tbp*). For analysis of relative expression levels for each target gene a replicate of the control group at 3dpf was set to one. The white, grey, and black bars in each graph represent expression levels for control, 2 µM $E_2$ treated and 5 µM $E_2$ treated groups respectively. Statistical differences of each treatment group versus the others are shown in white, grey, and black circles representing higher expressed than control, 2 µM $E_2$ treated and 5 µM $E_2$ treated groups respectively ($P < 0.05$). Error bars represent standard deviation calculated from three biological replicates. Each biological replicate was made from a homogenate of 30 heads.

estradiol (10 µM) giving rise to major disruptions of chondrogenesis followed by severe morphological defects (*Fushimi et al., 2009*). In the same study, analysis of gene expression after high dose estrogen treatment was limited to a semi-quantitative method (*in situ* hybridization) and a few chondrogenic genes belonging to only one molecular pathway (*Fushimi et al., 2009*). We hypothesized that many other candidate genes would be involved and hence, in the present study, we sought to quantitatively assess the expression of genes that could play role in the subtle effects of estrogen on the development of the craniofacial skeleton in zebrafish larvae (*Cohen et al., 2014*). Since our expression analysis depended on accurate qPCR, a prior step of careful validation of reference genes was essential to acquire reliable results (*Bustin, 2000*; *Kubista et al., 2006*). An increasing number of stably expressed reference genes have been validated for qPCR studies in a variety of fish species (*Ahi et al., 2013*; *Fuentes et al., 2013*; *Liu et al., 2014*; *Altmann et al., 2015*; *Wang et al., 2015*), and

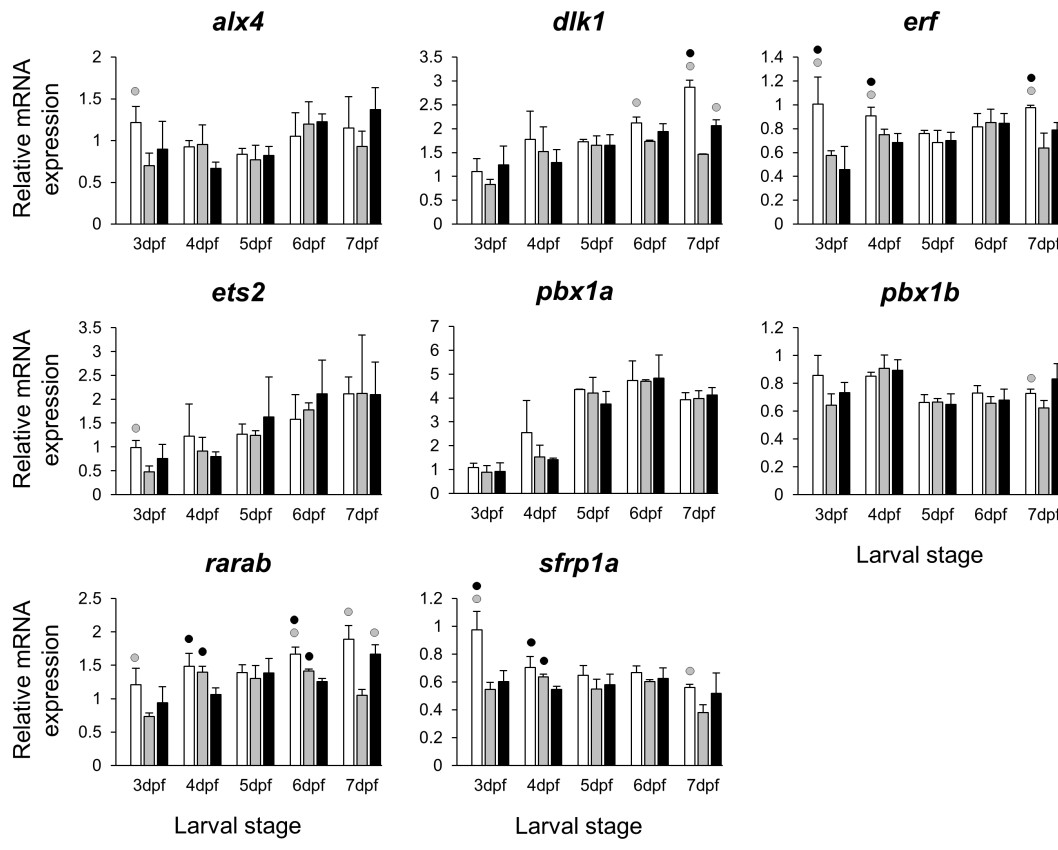

**Figure 5** **Expression differences of eight other potential targets of estrogen pathway involved in jaw skeletal elongation examined during zebrafish larval head development across control and $E_2$ treated groups.** Expression of *alx4*, *dlk1*, *erf*, *ets2*, *pbx1a*, *pbx1b*, *rarab* and *sfrp1a* was examined with qPCR and normalised using three best ranked reference genes (*ppia2*, *rpl8* and *tbp*). For analysis of relative expression levels for each target gene a replicate of the control group at 3dpf was set to one. The white, grey, and black bars in each graph represent expression levels for control, 2 µM $E_2$ treated and 5 µM $E_2$ treated groups respectively. Statistical differences of each treatment group versus the others are shown in white, grey, and black circles representing higher expressed than control, 2 µM $E_2$ treated and 5 µM $E_2$ treated groups respectively ($P < 0.05$). Error bars represent standard deviation calculated from three biological replicates. Each biological replicate was made from a homogenate of 30 heads.

also in zebrafish at different developmental stages, body parts/tissues, and treatments (*Tang et al., 2007*; *McCurley & Callard, 2008*; *Lin et al., 2009*; *Casadei et al., 2011*). There is however a necessity for validation of reference genes depending on the experimental conditions under study. Here, we found three genes, *ppia2*, *rpl8* and *tbp*, to be the most stably expressed candidate genes by all the methods of analysis used (Table 2 and Fig. 1) and their combination could ensure robust qPCR data normalisation (Fig. S1). We next selected candidate genes that are shown to be potential estrogen pathway targets, and at the same time, differential regulation of many of them is associated with morphological changes resembling shortened snout in different vertebrates (many found in mammalian species) (see publications referenced in Table 1). The underlying mechanisms by which these candidate genes could affect skeletogenesis are different from each other. For instance, genes like *bmp2a/b*, *rankl*, *runx2b* and *sox9b* are major factors in differentiation of skeletal

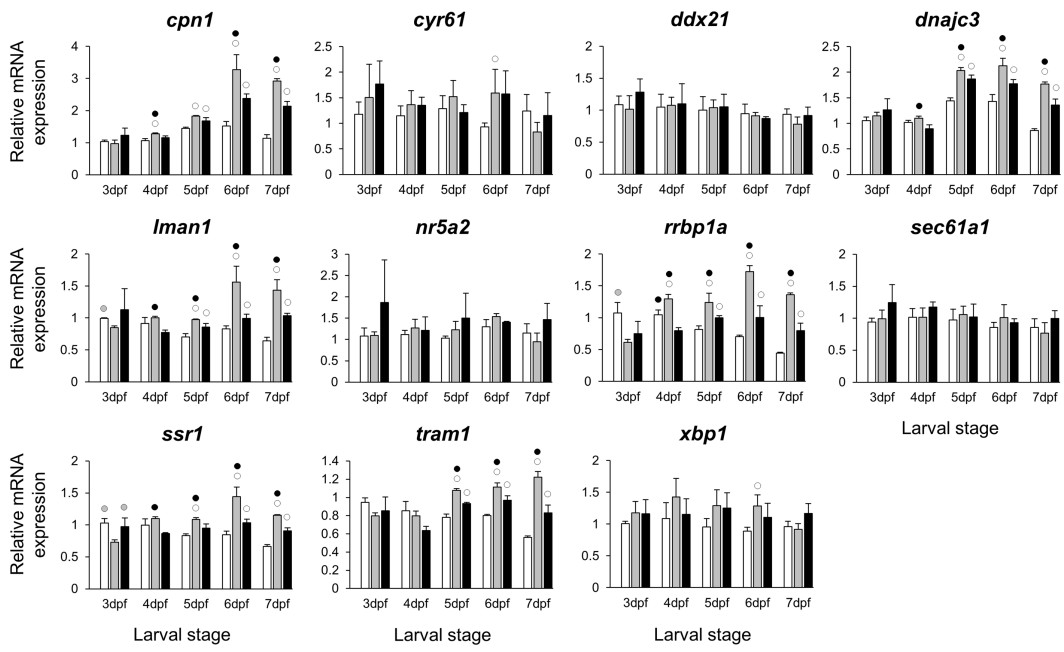

**Figure 6** Expression differences of *esr1* coexpressed genes in developing heads of zebrafish larvae across control and $E_2$ treated groups. Expression levels of eleven candidate genes coexpresed with *esr1*, based on data from COXPRESdb in zebrafish, were examined with qPCR and normalised using three best ranked reference genes (*ppia2*, *rpl8* and *tbp*). For analysis of relative expression levels for each target gene a replicate of the control group at 3dpf was set to one. The white, grey, and black bars in each graph represent expression levels for control, 2 µM $E_2$ treated and 5 µM $E_2$ treated groups respectively. Statistical differences of each treatment group versus the others are shown in white, grey, and black circles representing higher expressed than in control, 2 µM $E_2$ treated and 5 µM $E_2$ treated groups respectively ($P < 0.05$). Error bars represent standard deviation calculated from three biological replicates. Each biological replicate is based on a homogenate of 30 heads.

cells and some others such as *col2a1a*, *ctsk*, *mmp2/9/13*, *spp1* and *sparc* are critical for the formation of ECM in craniofacial skeletal structures (see Table 1).

The treatments with the two different doses of $E_2$ (2 and 5 µM) resulted in differential expression of many of the candidates during the zebrafish larval head development (Figs. 2, 3, 4 and 5). Consistent with a previous study in zebrafish using higher $E_2$ concentration (10 µM) (*Fushimi et al., 2009*), we also found significant down-regulation of *ptch1* and *ptch2* in the heads of fish receiving lower dose estrogen treatments during larval development. These two genes are the receptors (and the upstream mediators) of the hedgehog (Hh) signaling pathway which plays a crucial role in developmental patterning and skeletal morphogenesis (*Eberhart et al., 2006*; *Swartz et al., 2012*). Interestingly, slight changes in expression of *ptch1* were shown to be associated with subtle craniofacial skeletal divergence (shorter snout and flatter face) in cichlid fish (*Roberts et al., 2011*; *Hu & Albertson, 2014*). In addition, we found a strong positive expression correlation between *ptch1* and *ptch2* (Fig. S2), indicating potential estrogen mediated co-regulation of the two Hh receptors. In the above mentioned study of high dose $E_2$ treatment, the upstream activators of the Hh pathway, sonic hedegehog genes, *shha* and *shhb* (*twhh*), did not show significant changes in expression (*Fushimi et al., 2009*). However, this could be due to technical

limitations such as the use of a semi-quantitative method that is unable to reveal small differences in gene expression (*Fushimi et al., 2009*). In this study we found small and yet significant down-regulation of *shha*, but not *shhb*, in $E_2$ treated groups, as well as positive co-expression of only *shha* with the two Hh receptors. An important role of the *shh* in craniofacial skeletogenesis through activation of Hh signalling has been described (*Hu & Helms, 1999*), but it is not clear whether estrogen directly regulates its expression during development. The small reduction of *shha* transcripts in developing larval heads might be a result of a decreased number of cells expressing *shha* and not a direct estrogen mediated transcriptional regulation.

Extracellular matrix remodelling is a critical process in the developmental program of bone and cartilage differentiation and morphogenesis (*Werb & Chin, 1998*). The spatio-temporal expression of genes encoding matrix metalloproteinases and their tissue inhibitors plays a pivotal role in orchestrating the ECM remodelling process (*Werb & Chin, 1998*; *Page-McCaw, Ewald & Werb, 2007*). Moreover, many ECM remodelling genes are downstream targets of pathways mediated by nuclear receptors, including estrogen signalling (*Cox & Helvering, 2006*; *Heldring et al., 2007*; *Ganesan et al., 2008*). The selected ECM remodelling factors (*mmp2/9/13*, *timp2a* and *sparc*) were all reported to be regulated by estrogen signalling (*Lehane et al., 1999*; *Tüshaus et al., 2003*; *Marin-Castaño et al., 2003*; *Lu et al., 2006*; *Nilsson, Garvin & Dabrosin, 2007*; *Lam et al., 2009*; *Wang & Ma, 2012*) and play role in craniofacial skeletal morphogenesis (*Dew et al., 2000*; *Renn et al., 2006*; *Hillegass et al., 2007a*; *Hillegass et al., 2007b*; *Mosig et al., 2007*; *Rotllant et al., 2008*; *Letra et al., 2012*; *Ahi et al., 2014*). Our results revealed slight but significant effects of the estrogen treatments on expression of the selected ECM remodelling genes during larval head development (Fig. 4). It is interesting to note that previous investigations have shown association between differential expression of these genes and craniofacial phenotypes with flatter face and shorter snout (*Hillegass et al., 2007a*; *Hillegass et al., 2007b*; *Ahi et al., 2014*). The mechanism by which estrogen regulates the expression of ECM remodelling genes is not well understood. The estrogen dependent regulation might be exerted through interaction between estrogen-receptors and transcription factors that regulate ECM remodelling genes such as members of Ap-1 complex and ETS factors (*Lu et al., 2006*; *Ahi et al., 2014*; *Cao et al., 2015*). The binding motifs for Ap-1 and ETS transcription factors are present in the promoters of many ECM remodelling genes across vertebrate species (*Ahi et al., 2014*). Additionally, we found the expression of *erf*, an ETS repressor and estrogen target (*Sgouras et al., 1995*) to be down-regulated in both $E_2$ treated groups at three larval stages. Remarkably, a recent study showed that small reduction in expression of *erf* causes complex craniosynostosis with shortened snout in both human and mice (*Frasor et al., 2003*; *Twigg et al., 2013*). The same study also demonstrated regulatory elements containing Ap-1, ETS and Runx motifs as preferential *erf* binding sites (*Twigg et al., 2013*). Taken together, the results of the present and previous studies suggest potential estrogen mediated regulation of ECM remodelling genes possibly through interaction with other transcription factors. Other estrogen mediated processes than direct transcriptional regulation cannot, however, be ruled out, as the slight changes in transcript levels of ECM related genes could be due to reduced proportion of skeletal cells expressing these genes in larval heads. It is also

important to emphasize that the selected ECM genes can be expressed in other tissues of the head (though at considerably lower levels), thus their expression differences in other tissues might affect the overall changes in expression.

The $E_2$ treatments caused small and variable repressive effects on expression of other selected target genes (Figs. 2, 3, 4 and 5). The genes, *bmp2a* and *rankl*, are well characterized skeletogenic markers (*Nie, Luukko & Kettunen, 2006*; *Hu, Colnot & Marcucio, 2008*; *Lézot et al., 2015*) and their regulation by estrogen signalling has been reported in other vertebrate species (*Bord et al., 2003*; *Zhou et al., 2003*). It has been shown that treatment with high doses of $E_2$ can reduce the number of skeletal cells in the craniofacial skeleton (*Cohen et al., 2014*), hence the small changes in transcript levels of skeletogenic markers (e.g., *sox9b*) may again be caused by a decreased proportion of skeletal cells in the heads. We also found components of retinoic acid and Wnt/$\beta$-catenin signalling patways, *rarab* and *sfrp1a*, to be transcriptionally affected by $E_2$ treatment indicating the potential crosstalk of these pathways with estrogen signalling during larval head development (*Lohnes et al., 1994*; *O'Lone et al., 2004*; *Trevant et al., 2008*; *Yokota et al., 2008*). Although, the selected components of the pathways and transcription factors in this study (Fig. 5) are known to have markedly high levels of expression in the craniofacial skeleton, they might also be expressed to a lesser extent in other tissues within the larval head. Therefore, the observed small changes in expression can not be readily attributed to viscerocranial skeletal elements and further gene expression studies using dissected skeletal elements are essential to confirm this.

In addition to skeletogenic genes, we were interested in investigating the effects of different doses of $E_2$ on the expression of estrogen receptors. Therefore, we assessed the expression of two estrogen receptors, *esrra* and *esr1*, that could mediate estrogen signal during the development of skeletal tissues (*Bonnelye & Aubin, 2005*; *Bonnelye et al., 2007*; *Auld et al., 2012*). While the $E_2$ treatments had small and variable repressive effects on expression of *esrra*, the increased expression of *esr1* was observed in both $E_2$ treated groups. Strikingly, the lower $E_2$ concentration (2 μM) resulted in higher induction of *esr1* expression. This suggests that the distinct effects of lower doses of estrogen on craniofacial skeletogenesis, described by *Cohen et al. (2014)*, might be mediated by *esr1*, however further functional studies are required to demonstrate such a role. To identify genes sharing regulatory mechanisms in response to slight increases in estrogen levels, we further explored the expression of 11 genes constructing a co-expression network with *esr1* (Table S2 and Fig. 6). These candidate genes were selected by using a vertebrate co-expression database (*Obayashi & Kinoshita, 2011*) which we have successfully used for identification of gene networks associated with subtle craniofacial morphological divergence in another teleost (*Ahi et al., 2014*; *Ahi et al., 2015*). Our results indicate higher transcriptional induction of six genes, i.e., *cpn1*, *dnajc3*, *lman1*, *rrbp1a*, *ssr1* and *tram1* in the lower (2 μM), than the moderate (5 μM) treatment groups, during craniofacial development. The genes also showed positive expression correlation with *esr1* suggesting a common regulatory mechanism mediated by estrogen during head development. To our knowledge, a mechanism by which a lower concentration of estrogen can have stronger inductive effects on expression of certain genes than higher concentrations is not known.

Such a mechanism might be involved in distinct regulation of estrogen receptors by different concentrations of estrogen hormone, which in turn could lead to recruitment of the receptors to distinct genomic binding sites and/or with different binding affinity (*Stender et al., 2010*). Among the six genes only *dnajc3*, a gene encoding protein kinase inhibitor P58 (P58$^{IPK}$), has been shown to be involved in skeletogenesis through regulation of a cytokine-dependent cartilage degradation (*Gilbert et al., 2014*). Although all of the six genes have recorded developmental expression patterns in zebrafish craniofacial elements based on data in the ZFIN zebrafish database (*Thisse et al., 2001*; *Thisse et al., 2004*), their roles in craniofacial morphogenesis have yet to be investigated. Finally, an unbiased approach such as transcriptome sequencing rather than candidate gene-based study would be warranted to provide better knowledge of estrogen mediated effects on expression of genes with unknown roles in craniofacial morphogenesis as well as links between already identified genes and molecular pathways involved.

## CONCLUSIONS

In this study we quantitatively assessed the effects of two doses of estrogen (2 μM and 5 μM) on gene expression during zebrafish larval head development. We performed a highly sensitive and specific qPCR analysis and carefully validated reference genes. We assessed the expression of a selected set of genes involved in craniofacial skeletal development as well as genes coexpressed with *esr1*, an estrogen receptor showing stronger inductive response to 2 μM than 5 μM estrogen concentration. The results implicate estrogen in the expressional regulation of genes belonging to distinct signalling pathways such as hedgehog and retinoic acid pathways, as well as genes involved in ECM remodelling during craniofacial development. Furthermore, estrogen mediated transcriptional changes in a few tested major skeletogenic factors (e.g., *bmp2a* and *rankl*), and a transcription factor, *erf*, with a demonstrated role in the formation of a shortened snout phenotype in human and mouse. Finally, we identified a gene network showing positive expression correlation with *esr1* and higher induction in response to treatment with 2 μM than with 5 μM estrogen. This could suggest a co-regulated module of genes mediating the effects of low doses of estrogen during craniofacial development which is required to be further investigated at functional level.

## ACKNOWLEDGEMENTS

The authors would like to acknowledge Rebecca Hudon, Sean Ryan, and Alexander Kramer for assistance with zebrafish husbandry and sample isolation.

### Funding

This project was supported by The University of Iceland Research Fund, the Eimskip University Fund and Roanoke College, USA. The funders had no role in study design, data collection and analysis, decision to publish, or preparation of the manuscript.

## Grant Disclosures

The following grant information was disclosed by the authors:
The University of Iceland Research Fund.
The Eimskip University Fund and Roanoke College, USA.

## Competing Interests

The authors declare there are no competing interests.

## Author Contributions

- Ehsan Pashay Ahi conceived and designed the experiments, performed the experiments, analyzed the data, contributed reagents/materials/analysis tools, wrote the paper, prepared figures and/or tables, reviewed drafts of the paper.
- Benjamin S. Walker conceived and designed the experiments, performed the experiments, contributed reagents/materials/analysis tools, wrote the paper, reviewed drafts of the paper.
- Christopher S. Lassiter contributed reagents/materials/analysis tools, wrote the paper, reviewed drafts of the paper.
- Zophonías O. Jónsson conceived and designed the experiments, contributed reagents/materials/analysis tools, wrote the paper, reviewed drafts of the paper.

## Animal Ethics

The following information was supplied relating to ethical approvals (i.e., approving body and any reference numbers):

Zebrafish experiments were performed under the Roanoke College IRB protocol # 14BIO76.

## Data Availability

The raw data has been supplied as Data S1.

## Supplemental Information

Supplemental information for this article can be found online at http://dx.doi.org/10.7717/peerj.1878#supplemental-information.

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
