# Peer review of "Investigation of the effects of estrogen on skeletal gene expression during zebrafish larval head development"

_PeerJ, doi:10.7717/peerj.1878_

## Round 0.1 · original submission · Major Revisions

Both reviewers have recommended revisions to your paper. These revisions largely concern the way you have interpreted the data presented, rather than the way the experiments have been conducted. Accordingly, I ask that you take their comments seriously and provide a considered response. For your manuscript to be accepted, you may either come back with a satisfactory rebuttal for why you think all your conclusions are valid, or you must revise your interpretation and conclusions in several places, as recommended by the reviewers. I have marked the manuscript as requiring major revision so that you are aware that the requested modifications are important and that the manuscript requires quite substantial re-writing in parts. Please note, however, that although Referee 2 mentions that RNAseq analysis would have provided an unbiased and more comprehensive analysis of E2 target genes, we do not expect you to conduct this experiment.

Reviewer 1 ·

Basic reporting

The introduction and background were interesting and gave me an appropriate context for why identifying estrogen-dependent genes would be important.

Experimental design

The q-RT-PCR was done with appropriate controls, including internal control genes that are constant across stages and experimental conditions.

Validity of the findings

In the Cohen et al 2014 study, it was shown that higher levels of E2 result in markedly less cartilage in the zebrafish face. Thus, examination of whole heads at this higher dose may lead to skewed results if the number of cartilage cells per total tissue is less in the high E2 condition versus control. In other words, a reduction of cartilage genes (e.g. sox9) could be attributed to either regulation of sox9 transcription, or alternatively just to the fact that cartilage cells now represent a smaller fraction of overall cells. The discussion should acknowledge this possibility. In particular, the finding that a number of genes are found to be slightly decreased by E2 could represent such an indirect effect of cel proportions. While I found that increase in esr1 and co-regulated genes to be strongly supported, the variable and minor decreases of other genes were not as clear. For example, cartilage/skeletal differentiation is not much different between 4-6 days, so why should certain genes be slightly reduced at only a subset of these stages? I think a more conservative assessment of these slight expression changes should be made, especially when consistent changes are not seen across multiple stages. Also, sox9a and not sox9b is the major gene implicated in chondrogenesis in zebrafish, so I don't think analysis of sox9b is appropriate.

Reviewer 2 ·

Basic reporting

In this manuscript, the authors carefully validated reference genes for RT-qPCR during larval development and examined expression profiles of 39 genes in the dissected zebrafish larval head.

Experimental design

Major points

1. The authors claim that they identified E2 regulated genes during the development of craniofacial skeletal structures. One caveat to the accuracy of their results is that the isolated zebrafish head contains brain, eyes, ears, muscle, and possibly heart depending on how reproducibly the dissection was done, in addition to craniofacial skeletal structures. It is likely that at least some of the selected candidate genes are also expressed in one or more of the tissues mentioned above, in addition to craniofacial skeletal structures. Therefore the measured expression profiles for the 39 genes may not correspond perfectly to those in the craniofacial skeletal structures. It is possible that some of the smaller changes in gene expression that are observed, while being statistically significant, may be due to inconsistencies between samples. This should be mentioned in the Discussion. Instead of using the entire head, the study would have been superior if dissected pharyngeal cartilages had been used.

2. Instead of using a candidate gene RT-qPCR approach, RNA-seq experiments would have identified estrogen-mediated genes in an unbiased and more comprehensive fashion. Again, this should be mentioned in the Discussion.

Validity of the findings

3. The authors highlighted genes with a stronger inductive response to the lower dose (2 micromolar) of E2, but didn’t discuss the possible mechanisms for this. Was the intermediate level of E2 (5 micromolar) thought to be toxic?

Additional comments

Minor points

1. Line 143, a quick search on the COXPRSESdb came up with 338 candidate genes, not 339 as shown in the manuscript.

2. Line 144 and Table S1 (qPCR Primer sequences and information about esr1 co-expressed genes) cpn1 and cyr61 are not on the co-expressed gene list of 338 candidates.

3. Line 223, only 27 (not 28 as stated in the manuscript) genes were listed in Table 1

4. Line 231, spp1 is in Fig4, but not listed in the manuscript

5. Prepositions “in” and “at” appear interchangeably as follows: “in the last three stages”, “at the last stages” several times. Perhaps use “at the last stages” for consistency.

---

## Round 0.2 · accepted · Accept

Your revisions to the text of the manuscript are satisfactory and your responses to all the matters raised by the reviewers are adequate.